# Design of Gas Monitoring Terminal Based on Quadrotor UAV

**DOI:** 10.3390/s22145350

**Published:** 2022-07-18

**Authors:** Yang Liu, Lei Chen, Shurui Fan, Yan Zhang

**Affiliations:** Tianjin Key Laboratory of Electronic Materials Devices, School of Electronics and Information Engineering, Hebei University of Technology, Tianjin 300401, China; 202121902026@stu.hebut.edu.cn (Y.L.); 201931903018@stu.hebut.edu.cn (L.C.); zhangyan@hebut.edu.cn (Y.Z.)

**Keywords:** UAV, atmospheric environment, sensor calibration, monitoring terminal, cavity design

## Abstract

The problem of air pollution is an increasingly serious worldwide. Therefore, in order to better monitor the gas components in the atmosphere, the design of a gas monitoring terminal based on a quadrotor UAV, including software and hardware design, is hereby carried out. Besides, a pump-suction series cavity is designed to reduce the influence of airflow disturbance on the UAV, which is verified to possess a certain anti-interference ability through Computational Fluid Dynamics(CFD) simulation experiments. In addition, a linear regression algorithm is used for sensor calibration and a polynomial piecewise regression method is used for temperature compensation. The experimental results show that the R^2^ of the model reaches 0.9981, the fitting degree is rather high, and the output is closer to the real gas concentration value after calibration. At the same time, the temperature compensation parameters are determined, which considerably improves the accuracy of the entire hardware terminal. Finally, the vehicle exhaust monitoring experiment is conducted, and the experimental results show that this scheme can successfully detect the exhaust position of the vehicle exhaust under the interference of the downwash flow of the UAV, thereby proving the reliability and accuracy of the monitoring terminal.

## 1. Introduction

The atmospheric environment is one of the most important components of the natural environment, it is also the foundation for all living things in the world [1]. With the continuous development of modern urbanization, the problem of air pollution is increasingly worsened worldwide, and air quality directly affects the physical and mental health of contemporary people [2]. Long-term exposure to severely polluted air increases the incidence of cardiovascular and respiratory diseases, also lung cancer. According to statistics, a total of 4.9 million people died directly or indirectly from air pollution in 2017 [3]. In the case of comprehensive monitoring of air pollutants, air quality monitoring sites or portable air quality monitoring instruments are frequently adopted by many countries. However, the construction of the former requires a lot of supporting facilities and financial backup, which is also exposed to problems including a long construction period, low coverage density, and poor flexibility, while the latter can only monitor air quality in a small area. A large monitoring range requires a lot of labor costs. Besides, the effect is rather limited in some special scenarios, such as toxic and harmful gas leakage, where emergency personnel cannot intervene [4].

In recent years, the design of monitoring terminals for the atmospheric environment has attracted extensive attention. Atmospheric environment monitoring is the basis of atmospheric environment management and protection. Only by comprehensively, accurately and specifically understanding the atmospheric environment can effective prevention and control measures be implemented [5]. From the current domestic and foreign environment, the wired environment monitoring technology is relatively mature, but its application is still exposed to obvious limitations, including the application scope [6,7,8], making it necessary to design a system that can wirelessly transmit the collected ambient gas data to the host for processing [9,10]. Collecting data with a variety of sensors is a hot research topic at home and abroad, and related research has been carried out in many universities and research institutes [11]. Some scholars have proposed an atmospheric monitoring system composed of monitoring equipment and monitoring terminals, which sense ambient temperature, ambient relative humidity, ambient air pressure, carbon monoxide levels, carbon dioxide levels, and dust particle levels using sensors. This atmospheric monitoring system is wireless, in real-time, reliable, continuously recording, and portable [12]. Feng Zhenqiang proposed the design of a wireless sensor-based environmental monitoring system capable of the remote monitoring of temperature, humidity, air pressure, and PM2.5 data [13]; Philips has been working on the research and development of environmental monitoring and purification equipment [14], which can monitor and predict the atmospheric environment in real-time and is greatly convenient for managers; L. Capezuto put forward a monitoring method that allows people to obtain their own sensor nodes and participate in aerial environment monitoring through the mobile terminal [15]; Popoola constructed a sensor network platform composed of low-cost air quality sensors for quantifying air quality in urban environments [16]. However, in the future of rapid development, mobile terminals have been widely used in actual production and daily life [17], such as UAVs for intelligent agricultural irrigation [18] and real-time traffic monitoring [19] to strengthen the support of the environmental protection industry and accelerate the improvement of the technical equipment level of the environmental protection industry. In addition, Pretto et al. from the Swiss Federal Institute of Technology in Zurich carried out research on the use of UAV systems for fine planting and proposed a plant protection scheme for aerial monitoring of vegetation using rotary-wing UAVs, which greatly reduces labor costs [20]. Some scholars have created a learning-based UAV system for autonomous surveillance, in which UAVs can be used to autonomously detect and track target objects without human intervention [21]. The premise of preventing and controlling air pollution is to obtain information on the pollution degree. UAVs can be used as the carrier of atmospheric environment monitoring devices for the collection of pollution information in three-dimensional space for specific polluted areas, such as industrial parks and pollution treatment plants. This very research direction has become a new research hotspot [22,23].

Low-cost sensors (LCSs) have emerged in recent years as complementary tools that, combined with conventional equipment, allow air quality to be monitored more effectively, dramatically improving the spatial resolution of air-monitoring data and effectively engaging citizens in the experimental measurements [24]. In [25], a low-cost sensor (LCS) was used for monitoring the atmospheric environment with high spatial resolution, and an Atmospheric Exposure Low-Cost Monitoring (AELCM) system for several air species was proposed, including ozone, nitrogen dioxide, carbon monoxide and particulate matter, as well as meteorological variables developed by our research group, but only at local scales. Due to the low-cost advantage of commercial sensors, they are used as the key components of IoT devices in atmospheric state monitoring, and an integrated machine learning model is used to detect accurate concentrations [26]. In [27], a low-cost electrochemical and optical sensor-based device was proposed, attached to the handlebars of bicycles for monitoring air quality in urban environments. The system consists of three electrochemical sensors for measuring NO_2_ and O_3_ and an optical particulate matter (PM) sensor for PM_2.5_. This article mainly focuses on the development and calibration of the rotor-borne atmospheric gas monitoring terminal. Rotorcrafts are more flexible than fixed wing in maneuverability and are more suitable for air environment monitoring [28]. Firstly, the hardware scheme is hereby designed for the monitoring terminal and a variety of sensors are selected. Given that the downwash air flow generated by the UAV during the movement will interfere with the final reading of the monitoring terminal, a pump-suction series cavity is designed to solve this problem; it is proven to possess a certain anti-interference performance through computational fluid dynamics simulation(CFD) and experimental verification. The electrochemical gas sensor is calibrated using the linear regression algorithm, while polynomial regression is used for temperature compensation. Additionally, the software design is completed, and a vehicle exhaust gas detection experiment is carried out to verify the feasibility of the system.

The hardware scheme design in Section 2.1, includes sensor selection and hardware circuit design. In Section 2.2, an introduction to the design of the tandem cavity is carried out. The need for sensor calibration and temperature compensation methods is described in Section 2.3. Moreover, the design of the software scheme is carried out in Section 2.4. Next, the CFD simulation experiment is carried out in Section 3.1. The sensor calibration experiment is carried out in Section 3.2. Moreover, the segmented temperature compensation method experiment is carried out in Section 3.3. Finally, the vehicle exhaust gas detection experiment is carried out and the conclusion is drawn.

## 2. Materials and Methods

### 2.1. Hardware Solution Design

In this chapter, the software and hardware scheme is designed first, followed by the series cavity design and CFD simulation. Finally, the linear calibration method and the segmented temperature compensation method are proposed for the errors generated by the electrochemical sensor.

The hardware structure of the monitoring terminal consisting of a sampling unit and a detection unit is shown in Figure 1, the sampling unit, includes a micro brushless air pump, a gas sensing array, a particle sensor, and a temperature and humidity sensor. The process is that the air pump continuously pumps the outside air to the gas sensing array to generate a response signal, and meanwhile, the particle sensor detects PM_10_ concentration in the air, while the temperature and humidity sensor detects the temperature and humidity of the location of the UAV in real-time. Besides, the detection unit contains a power battery, a core processor, a data storage module, and a communication interface, among which, the power battery is used for driving the entire system; the communication interface is used for external communication, and data storage is used for data storage. Furthermore, the core processor is used for driving each sensor to work normally, collect sensor data, then preprocess and store the sensor data. Finally, it is transmitted through the communication interface.

#### 2.1.1. Sensor Selection

The monitoring targets of the hereby designed monitoring terminal are various air pollution chemical components, inhalable particulate matter, and temperature and humidity. For this purpose, a sensor array composed of temperature and humidity sensors, electrochemical gas sensors, particulate matter sensors and CO_2_ sensors is selected.

(i).Electrochemical gas sensors

Electrochemical gas sensors are provided with the advantages of high selectivity, strong linearity and low power consumption [29], and common electrochemical sensors include the two-electrode type, three-electrode type, and four-electrode type. The two-electrode type has a relatively simple structure and only connects to the working electrode and the counter electrode through a load resistance. In the case of a slightly higher measured gas concentration, the potential of the sensitive electrode will exceed its allowable range, so that the output is no longer linear. Besides, its upper limit of detection is low, so it is not selected. The three-electrode type introduces a reference electrode, which greatly improves the detection range, and its structure is shown in Figure 2. The four-electrode type adds an auxiliary electrode on the basis of the three-electrode type to improve the selectivity and eliminate the influence of temperature and humidity, but the cost is much higher and the basic driving circuit is more complicated, so it is not selected either.

There are many types of three-electrode electrochemical sensors produced by Honeywell in the United States of uniform size, which is convenient for replacement while monitoring the target gas changes in the future. However, Honeywell does not produce ozone sensors of the same size. In this case, the three-electrode ozone electrochemical sensor produced by Shengmi Technology Company is selected. Some parameters of each sensor are shown in Table 1.

(ii).Particulate sensor

For the detection of inhalable particles, the laser sensor based on the principle of light scattering is currently the most widely used. The light scattering particle sensor is produced using the Mie scattering theory. Scattering occurs when light is irradiated on suspended particles, and the mass and concentration will be calculated according to the scattering characteristics to achieve the conversion of optical signals to electrical signals, which possess the characteristics of simple structure, fast response speed and high precision. The performance of various commercial particulate matter sensors is compared and analyzed, as shown in Table 2.

(iii).CO_2_ sensor

At present, there are many kinds of CO_2_ sensing technologies, such as non-dispersed infrared (Non-Dispersed Infrared Red, NDIR) technology, gas chromatography, electrochemical method, etc., among which, gas chromatography detects carbon dioxide after being completely separated from other components of air in the chromatographic column. However, the chromatographic analysis conditions often vary under different experimental conditions. Besides, standard gas needs to be configured during the measurement process, and pure nitrogen is generally used for the standard gas, which not only undergoes a cumbersome process but also requires a long operation period; the electrochemical method has high sensitivity, good accuracy, simple instrument but poor selectivity; NDIR technology possesses good selectivity and a fast response speed for CO_2_ detection [30,31], and is mostly used in indoor environmental monitoring, industrial production emission monitoring and other scenarios [32,33]. Based on the above comparison, this article chooses NDIR technology and compares a variety of CO_2_ sensors using this technology as shown in Table 3.

S8-0053, SCD30 and DS-CO_2_-20 are all provided with high monitoring accuracy and fast response speed. However, given that the monitoring terminal needs to be mounted on the UAV to operate smoothly, the airflow interference generated by the UAV will lead to lower monitoring accuracy. T6615-5KF is an inhalation detection method, which is more capable of resisting the impact of air pressure changes, though with a relatively poor monitoring accuracy. In addition, it is also suitable for the airborne application of UAVs. In this case, T6615-5KF is finally selected.

(iv).Temperature and humidity sensor

The hereby selected temperature and humidity sensor is SHT21 which uses the I2C communication mode, the temperature monitoring range is −40~125 °C; the temperature monitoring accuracy, 0.3 °C; the relative humidity monitoring range, 0~100%; the relative humidity monitoring accuracy, 2% and the response time, 2 s.

#### 2.1.2. Hardware Circuit Design

The STM32F429VGT6 microprocessor launched by ST Company is hereby selected as the core processor of the main control unit. This processor is based on the ARM Cortex M4 core, with fast data processing speed and large storage space and is, therefore, widely used in embedded devices [34]. It has eight channels of serial ports, four channels of SPI and three channels of I2C, rich I/O resources, and peripheral circuits including a reset circuit, a crystal oscillator circuit, a status indicator, and an RTC power supply.

The output of the three-electrode electrochemical sensor is a weak current signal with a classic conditioning circuit, and its structure is shown in Figure 3. A potentiostat circuit is formed by operational amplifiers U1, U2, and external reference sources $V_{ce}$ and $V_{mid}$ to meet the fixed bias voltage between the working electrode (WE) and the reference electrode (RE) of the three-electrode electrochemical sensor mentioned above, that is, the amplifier U2 and feedback resistor $R_{f}$ form an I/V conversion circuit. The formula for calculating the magnification can be expressed as:(1)Vout=Vmid−IinRf
where $I_{in}$ denotes the current output by the working electrode (WE) terminal, and $V_{out}$ represents the output of the operational amplifier U2.

The classic conditioning circuit is needed to modify the feedback resistance and the reference source voltage and can drive different sensors normally. However, its adaptability is poor, the structure is complex, and it is difficult to stably maintain the constant potential of the sensor with specific bias requirements. Therefore, the LMP91000 electrochemical sensing front-end chip is hereby adopted as the processing core of the output signal of the electrochemical sensor. The working principle of the LMP91000 is shown in Figure 4.

According to Figure 4, the Vref voltage divider controls the potential of the “+” terminal of the TIA amplifier and the “+” terminal of the A1 amplifier by programming. According to the virtual short principle of the operational amplifier, a fixed bias voltage can be realized between WE and RE. In addition, the resistance values of the load resistance and feedback resistance are also selected by programming. For the WE electrode current outflow type sensor of the electrochemical sensor, the weak output current signal can output an mV level voltage signal after passing through the TIA amplifier. The conversion relationship is shown in Formula (2):(2)$$V_{out}=V_{TIA}-I_{in}R_{tia}$$

For the WE electrode current inflow type sensor of the electrochemical sensor, the conversion relationship is shown in Formula (3):(3)$$V_{out}=V_{TIA-}+I_{in}R_{tia}$$
where $V_{TIA-}$ is the potential of the “−” terminal of the transimpedance amplifier TIA; $V_{out}$, the output voltage, and $I_{in}$, the current signal output by the working electrode.

The load resistance matters considerably in reducing the noise of the sensor output, but a high load will extend the response time and polarization time. Therefore, the selection of $R_{load}$ needs to achieve a good balance between low noise and response speed, generally within the range of 15~50 Ω.

The circuit connection between the electrochemical sensor and the LMP91000 is shown in Figure 5, where UA1 is an electrochemical sensor, and UA2 is an LMP91000. The electrochemical sensor requires a long polarization time to maintain its stable working state after being powered on. In addition, in order to keep the electrochemical sensor in the “preparatory work” state when the circuit is disconnected, it is necessary to short-circuit the working electrode and the reference electrode pin during the disconnection. More importantly, it is of great significance to disconnect the short circuit between the two pins in time when powering on. For the above reasons, a P-channel JFET (Q1) is thus introduced between the two electrodes. During the power-off period, the D-poles are turned on to achieve the short-circuiting effect, and they are disconnected immediately after power-on.

The LMP91000 communicates with the STM32 microprocessor through the I2C protocol. Both SCL and SDA pins are connected to the same pair of I2C interfaces of the microprocessor. LMP91000 has a chip select pin MENB. By controlling the level of this pin to determine whether it is in the selected state, multiple LMP91000s are enabled to communicate with the microprocessor. The structure is shown in Figure 6. The LMP91000 electrochemical sensing front-end chip greatly reduces the complexity of circuit design. Furthermore, corresponding parameters can be configured according to the characteristics of different electrochemical sensors, which thereby improves adaptability.

The filtering and signal acquisition circuit consists of voltage follower UA3, RC filtering and 24−bit high-precision A/D acquisition module AD7793, as shown in Figure 7, among which, AD7793 is a 24−bit three−channel low−noise differential analog input ADC produced by ADI, with a sampling rate of 4.17 Hz~470 Hz. It also integrates an on-chip low−noise instrument op amp programmed and controlled by SPI. The microprocessor can communicate data with multiple AD7793 through a chip select pin, and the electrochemical sensor is input to the voltage follower after signal processing. The voltage follower has the characteristics of high input impedance, low output impedance, and a gain of 1, which hardly interferes with the signal itself, and can effectively suppress the influence of the electrical characteristics between the front and rear stages.

The ADR441 reference source chip is used to provide the Vref+ of 2.5 V for AD7793 and LMP91000, and has the characteristics of high precision, low noise and low drift, and an output error of only ±3 mV. In addition, given that the output signal of the electrochemical sensor is weak and susceptible to interference, the digital−analog isolation modules ADuM1401 and ADuM1400 are used for isolating the SPI signal and the chip−select signal, respectively.

The CO sensor is taken as an example for signal analysis according to the electrochemical sensors selected in Table 1, which has a measurement of 0–500 ppm, a sensitivity coefficient of 70 ± 15 nA/ppm, and a bias voltage of 0 mV. Assuming that the sensitivity is 70 nA/ppm, the maximum current output becomes 35 uA according to the range. At the same time, the Vref divider of LMP91000 is controlled to maintain the bias between the RE and WE at 0 mV. According to Formula (1), the larger the $V_{TIA-}$ is, the better the amplification effect becomes. Select 67% of the maximum reference source, and the amplified signal is required to be lower than 1675 mV, so the value of the signal is within the range of A/D. Therefore, $R_{tia}$ is configured accordingly to 35 kΩ. According to Formula (2), the value of the amplified A/D sampling signal is calculated as 450 mV.

As the core driver of the whole system, the power supply module supplies power to each module, and its stability and load capacity have a huge impact on the performance of the entire system. In order to ensure the power supply performance, the power supply module is designed as shown in Figure 8, which is powered by two 3.7 V lithium batteries in series. Since the working voltage of the dust sensor and other modules is 5 V, and the working voltage of the main control chip is 3.3 V, the SPX29300-3V3 and SPX29300-5V high-power linear voltage regulator chips are selected to provide stable voltages of 5 V and 3.3 V. The input pins of the two voltage regulator chips are connected to 330 uF electrolytic capacitors for energy storage and effectively reduce the power input ripple. Considering the poor characteristics of large capacitors in filtering high-frequency signals, a 0.1 uF filter is connected in parallel at the back end. The capacitor also constitutes a secondary filter, and the capacitor introduced at the output end of the voltage stabilizer can effectively suppress the ripple generated by the voltage conversion of the voltage stabilizer. The magnetic beads are connected to the circuit in series to isolate the 3.3 V digital signal power supply from the analog signal power supply. In order to protect the circuit safety, a one-time fuse F2, Schottky diode D7, and TVS tube D8 are added to the input end of the power supply to prevent circuit overcurrent, reverse connection and absorb surges. At the same time, a micro-USB power supply interface is designed to meet the requirements of electrochemical sensor laboratory testing and preheating before use. The power input is filtered by a capacitor to provide a stable voltage of 5 V and input to the LM1117S-3.3 linear voltage regulator module, so a stable voltage of 3.3 V is provided by the LM1117S-3.3, and the front and rear capacitors.

The rationality of the PCB layout, line width, component spacing, power supply and bottom line routing, are all the keys to the overall performance of the realization of the hardware circuit. The physical diagram of the PCB is shown in Figure 9.

### 2.2. Design and Simulation Analysis of Pump Suction Series Cavity

During the flight and hovering of the UAV, the air velocity and gas concentration distribution around the fuselage is affected by the downwash flow of the propeller, which leads to certain errors in the gas sensor [35]. In order to minimize the error and improve the data accuracy, the air pump is used to send the external gas into the cavity, and the gas response cavity is thereby designed.

According to the size of the electrochemical sensor mentioned above, two cavity structures are designed as shown in Figure 10. CFD is an emerging discipline that simulates and analyzes fluid mechanics using computer and numerical methods. The powerful computing power of CFD provides technical support for fluid mechanics. In this case, a CFD simulation is hereby used for simulating the distribution of the gas flow field and pressure field when the gas flows through the cavity, which can effectively evaluate the influence of different cavity structure designs on the response of the electrochemical sensors. Besides, it is also required to mesh the part of the cavity through which the gas flows after completing the 3D modeling of the cavity in CAD. In addition, after importing the divided model, it is rather essential to select the simulation material, fluid model, boundary conditions, air inlet and outlet and set the initial conditions of the airflow field. Finally, the numerical solution is completed by a discrete solver.

(i).Fluid model selection

The calculation model of CFD depends on the properties and flow state of the fluid. The Reynolds Number (*Re*) defines the ratio between the inertia and the viscous force of the fluid, and the calculation formula is shown in Formula (4). The fluid model can be determined by calculating Re: it is laminar flow when *Re* < 2300; transitional flow when *Re* = 2300~4000, and turbulent flow when *Re* > 4000.
(4)$$Re=\frac{\rho vd}{\eta },$$
where ρ is the fluid density and viscosity coefficient; *d*, the diameter of the tubular micro-element; v, the fluid flow velocity, and *η*, the fluid viscosity. The Reynolds number is calculated as greater than 4000, so the k−ε turbulence model is selected for airflow simulation.

(ii).Cavity meshing

It is necessary to simulate the flow field in the cavity by meshing the gas flow field. A comparison between the cover cavity and the cap cavity is shown in Table 4. Different divisions are made according to the characteristics of the two types of cavities. The cover cavity is shown in Figure 11a, and the cap cavity is displayed in Figure 11b.

(iii).Setting of boundary conditions

In the case of performing numerical simulation calculations in the cavity, it is necessary to set the boundary conditions, mainly including the environmental conditions, velocity inlet, and pressure outlet. For the environmental conditions, the type of airflow entering the flow field is air, and the temperature is set to 293.15 K. One side is selected as the inlet boundary. At the same time, the flow velocity is calculated according to the flow rate of the air pump and the size of the air inlet, and the normal inflow velocity of the velocity field of the inlet boundary is set to 2 m/s. Meanwhile, the other side serves as the outlet boundary, and the pressure is 1 atm.

(iv).Solving method

For the solution method, the finite element method is used to complete the solution based on the three basic conservation equations of flow, including the mass conservation Equation (5), the momentum conservation Equation (6), and the energy conservation Equation (7), expressed as follows:(5)$$\frac{\partial \rho }{\partial t}+\nabla \left(\rho u\right)+\nabla \left(\rho v\right)+\nabla \left(\rho w\right)=0$$
(6)$$\rho \frac{\partial \vec{v}}{\partial t}+\varphi \left(\vec{v}\nabla \right)\vec{u}=\nabla \left[-pI+\tau \right]+\vec{F}$$
(7)$$\rho C_{p}\left(\frac{\partial T}{\partial t}+\left(\vec{u}\nabla \right)\right)=-\left(\nabla \vec{q}\right)+\left(\tau :\nabla u\right)-{\left(\frac{\partial ln\rho }{\partial lnT}\right)}_{p}\frac{dp}{dt}$$
where ρ is the fluid density; x,y,z, three orthogonal vectors; *u*, *v*, *w*, the components of the velocity field in the x,y,z directions; *p*, the pressure; τ, the viscous stress; $\vec{F}$, the volume force; $C_{p}$, the specific heat capacity; T, the temperature, and q, the heat flux.

The solution results are shown in Figure 12, where (a) and (b) are the simulation results of the velocity field streamlines, and (c) and (d) are those of the pressure field. The following conclusions can be drawn from the simulation results: when the airflow flows into the two cavities, the airflow in the cavity will be seriously diffused and most of the airflow will flow straight to the outlet due to the large calculation area inside the cover cavity; there is very little flow at the top, which not only prolongs the response time of the sensor but also causes errors in the accuracy of the results. Therefore, the cover cavity will not achieve the expected results and the internal computing area of the cap cavity is small, and most of the airflow flows directly to the top of the sensor, which greatly shortens the response time of the sensor.

By comparing and analyzing the flow direction of the airflow inside the cover cavity and the cap cavity, and their influence on the results through the above simulation results, it can be concluded that the cap cavity structure is more suitable for the actual needs of rapid sampling. Therefore, a series cavity with a buffer cavity is hereby designed on the basis of the cap cavity structure as shown in Figure 13, which is also simulated, and the simulation results are shown in Figure 14.

It can be seen from the above simulation results that the airflow conditions in the response cavity of each small cavity are basically the same in the designed series cavity, and the designed buffer cavity can affect the airflow when the speed of the air pump fluctuates. The buffering function keeps the airflow in the response cavity relatively stable.

### 2.3. Sensor Calibration and Compensation Method

Considering the process error of electrochemical sensor fabrication, there are certain differences in the sensitivity and baseline drift of each sensor. In order to make the monitoring results more accurate, the sensor should be necessarily calibrated for a more accurate sensitivity and zero voltage [36]. The reaction basis of the electrochemical sensors is a chemical reaction, the response of which is easily affected by temperature changes, thereby resulting in output drift in the case of climate changes throughout the year. Therefore, temperature compensation for the electrochemical sensors is also required. To this end, a linear regression algorithm is adopted for sensor calibration and a polynomial piecewise regression method is chosen for temperature compensation.

### 2.4. Software Solution Design

In order to drive the electrochemical sensor to work properly and read the current value, the LMP91000 and AD7793 are configured as shown in Figure 15. The LMP91000 is initialized and its power-on status is read after power-on. A value of 1 means that it is ready to receive I2C commands, unlock the write function to the chip and configure the TIACN and REFCN registers for setting variables including the load impedance, transimpedance gain, internal zero point potential, bias voltage, reference source and working mode. Finally, it needs to be locked to prohibit writing to the register. Then the AD7793 is initialized, and at the same time, the microprocessor sends an instruction to read the correctness of the register ID. For a correct ID, the first step is to set the configuration register, i.e., to select the gain, reference source and differential mode input channel, while the second step is to set the mode register, i.e., to set the working mode, clock source and sampling frequency. Finally, the ADC conversion result can be read from the data register.

AD7793 selects the external reference source as the sampling reference, the sampling range of which is 0~2.5 V, and the gain is 1. Additionally, the differential input channel 2 and the internal clock source are selected, and the sampling frequency is set to 16.7 Hz. According to the range and sensitivity parameters of the four electrochemical sensors selected in Table 1, the LMP91000 sets the potential of the TIA “−” terminal to 67% of the reference source, that is, $V_{TIA-}$ = 1.675 V and the bias voltage is 0 mV. In order to make the output range of the electrochemical sensor after the amplifier within the A/D sampling range as far as possible, the values of $R_{tia}$ in the LMP91000 are selected as follows: CO sensor—350 kΩ; SO_2_ sensor—120 kΩ; NO_2_ sensor—120 kΩ, and O_3_ sensor—120 kΩ. Besides, the load resistance $R_{load}$ is selected as 50 Ω.

The STM32F429VGT6 processor reads the readings of each electrochemical gas sensor using the SPI chip selection polling method, and these values are calibrated through a calibration model and a temperature compensation model. Then, readings of the CO_2_ sensor, particulate matter sensor, and temperature and humidity sensor are read in turn. All data are stored in an SD card for backup, and the data are sent in UART mode. At the same time, warning lights are set up to remind the programmer of errors.

## 3. Results

In this chapter, firstly, the CO electrochemical sensor is taken as an example to conduct a CFD experiment on the series cavity designed in the previous chapter. Then, the parameter calibration experiment based on the linear regression model and the piecewise temperature compensation method experiment are, respectively, carried out to reduce the errors caused by the sensor.

### 3.1. Experimental Verification and Analysis of CFD Simulation

In order to verify the consistency of the simulation results with the actual effect, the CO sensor is taken as an example to carry out the verification experiment. The 3D printing technology is used to realize the serial cavity, and nylon material is chosen for the printing. Firstly, the cavity is installed on the CO sensor. When the sensor polarization is completed and the output is stable after 2 min of power-on, clean air is introduced into the cavity for 2.5 min. At the same time, the gas mixture is controlled by a Mass Flow Controller (MFC). Then, the air is mixed with 1000 ppm CO according to the ratio of 98:2, and passes the cavity for 2.5 min. Finally, clean air passes through the cavity for 2.5 min. Since the electrochemical sensor will generate a weak current itself when not in contact with the target gas, the output is considered the baseline current. Therefore, the sensor baseline is corrected following Formula (8) in the data processing process, where I is the current of the sensor after baseline processing; $I_{t}$, the original current value of the sensor at time *t*, and *N*, the sampling times, generally within the range of 60–120. Excluding the polarization process data, the experimental results after baseline processing are shown in Figure 16.
(8)$$I=It-\frac{1}{N}\sum_{t=1}^{N}It$$

The experimental results show that the response generated by the CO sensor is smooth and stable under the series cavity, and the time (t_90_) for reaching 90% of the stable response is about 30 s.

The monitoring terminal is mainly used for monitoring atmospheric environmental pollution. The concentration of various polluting gases is low in the atmospheric environment. Therefore, the measured gas is mixed to a low concentration in the follow-up experiment. In order to verify the series cavity’s capability of achieving the same effect, four CO electrochemical sensors of the same batch are installed on the series cavity for experiments, with the CO gas test as an example.

For a low measured gas concentration value, the sensor tends to exhibit relatively large noise, which requires the obtained data to be denoised. Exponentially Weighted Moving-Average (EWMA) is an algorithm that estimates the current value by combining it with historical observations, as shown in Equation (9) [37]. When the current value is predicted, the weight of the historical observation value decreases exponentially with time. The closer the data are to the current moment, the higher the correlation is and the greater the weight becomes, which can effectively reflect the recent change trend.
(9)$$Y_{i}={\left(1-k\right)}^{i}\rho_{0}+k{\left(1-k\right)}^{i-1}{\bar{X}}_{1}+k{\left(1-k\right)}^{i-2}{\bar{X}}_{2}+\cdots +k{\left(1-k\right)}^{i-(i-1)}{\bar{X}}_{i-1}+k{\left(1-k\right)}^{i-i}{\bar{X}}_{i},i>0,$$
where $\rho_{0}$ is the initial time value; $\rho_{0}$ =$\;Y_{0}$,${\bar{X}}_{i}$, the observed value at the time; *Y_i_*, the predicted value at the time and *k*, the weighted descending speed. The larger the value is, the faster the weighting decreases.

The coefficient can be expressed as the half-life $\tau_{half}$ function [38]:(10)$$k=1-\exp\left(\frac{\log0.5}{\tau_{half}\cdot f_{s}}\right)$$
where $f_{s}$ is the sampling frequency. The experimental results before and after EWMA filtering and the experimental results of four CO sensors at the same time are shown in Figure 17 and Figure 18, respectively.

In the figure, CO_1_EWMA, CO_2_EWMA, CO_3_EWMA, and CO_4_EWMA represent the response curves of the four CO sensors installed in the series cavity from left to right after EWMA filtering, respectively. The horizontal axis represents time, and the vertical axis is the output current value of the working electrode. The experimental results show that the response duration and response start time of the four CO sensors in the series cavity are basically the same, and there is almost no delay problem, which is consistent with the simulation results.

### 3.2. Sensor Calibration Experiment

The monitoring terminal and standard equipment are connected to the gas circuit in series. Similar to the above experimental process, clean air is introduced for 2.5 min when the polarization of each sensor is completed, and the MFC is controlled to introduce a certain concentration of a single type of target gas for 2.5 min. Then, the target gas concentration is increased through MFC and passed into the gas path. A gas concentration step is formed after several increases in the concentration. The current output value I of the monitoring terminal and the gas concentration value C are recorded during the whole process, and the experiment is repeated five times.

The response generated by the electrochemical sensor in its range has a linear relationship with the target gas concentration. Taking the CO sensor as an example, its sensitivity is 70 ± 15 nA/ppm according to Table 1. A linear model is built accordingly as shown in Formula (11), where $\hat{C}$ is the gas concentration predicted by the model; I, the current value output by the sensor; w, the sensitivity correction factor and *b*, the offset.
(11)$$\hat{C}=\left(70+w\right)I+b$$

The preprocessing and model training process is shown in Figure 19. The ratio of the training set to the test set is 7:3.

The data recorded by the monitoring terminal are processed by the baseline and filtered by EWMA. At the same time, the average values *I*_0_, *I*_1_, *I*_2_, …, *I_n_* (*n* > 0, *n*∈R) of the 1 min data are taken after each step is stabilized, and these average values are input into the linear model for prediction. The loss function is L1Loss as shown In Equation (12), which takes the absolute value of the difference between the true value of gas concentration measured by standard equipment and the model predicted value. Adam optimizer is used for parameter optimization, and the learning rate is 0.5.
(12)$$loss\left(x,y\right)=\frac{1}{n}\sum_{i=1}^{n}\left|y_{i}-{\hat{y}}_{i}\right|$$

During backpropagation, the parameters *θ* (including the sensitivity factor *w* and bias *b*) are updated by minimizing the loss product of the gradient *θ* and the learning rate *a* as shown in Equation (13). Backpropagation is automatically handled by the Adam optimizer under the pytorch framework. The linear regression model can better fit the training set data after each iteration of feedforward and backpropagation.
(13)$$\theta =\theta -a\cdot \frac{\partial loss}{\partial \theta }$$

After 150 iterations, the loss tends to be stable, and the training results of parameters *w* and *b* are *w* = 2.754541, and *b* = 0.4882. The performance in the test set is shown in Figure 20a, where it can be observed that R^2^ = 0.998106, and the fitting degree is rather high. The comparison of the linear regression model before and after calibration is shown in Figure 20b. After calibration, the output is closer to the real gas concentration value.

### 3.3. Design of Segmented Temperature Compensation Method

The temperature compensation design is hereby carried out according to the factory test temperature compensation curve in the data sheet of each sensor. Considering the strong nonlinearity of the temperature compensation curve, the order of the model must be much higher to achieve a better fitting effect. Additionally, the adoption of the linear regression method results in a segmentation phenomenon in the results. Therefore, in order to reduce the complexity of the model, piecewise polynomial regression is used for designing the temperature compensation as shown in Equation (14), where *C* denotes the gas concentration value after calibration; *C_T_*, the concentration value detected by the electrochemical sensor at the temperature *T*; *T*, the temperature, and the unit is °C; a, b and c the weight of temperature at order 1, 2 and 3, respectively, and d, the bias. The training results of each parameter are shown in Table 5. Figure 21 depicts the temperature compensation curves of various sensors corresponding to the model.
(14)$$C=\{\begin{array}{c} C_{T}/(a_{1}T+b_{1}T^{2}+c_{1}T^{3}+d_{1})\%,-10\;℃\leq T\leq 0\;℃;\\ C_{T}/(a_{2}T+b_{2}T^{2}+c_{2}T^{3}+d_{2})\%,0\;℃\leq T\leq 15\;℃;\\ \begin{array}{c} C_{T}/(a_{3}T+b_{3}T^{2}+c_{3}T^{3}+d_{3})\%,15\;℃\leq T\leq 30\;℃;\\ C_{T}/(a_{4}T+b_{4}T^{2}+c_{4}T^{3}+d_{4})\%,30\;℃\leq T\leq 45\;℃; \end{array} \end{array}$$

## 4. Discussion

So far, the design of the entire gas monitoring terminal has been completed. An automobile exhaust pollution point detection experiment is hereby designed to verify the scheme feasibility, and the F330 rotorcraft is used and the airborne monitoring terminal is also installed directly below the UAV to ensure the safety of the experiment. A fuel vehicle is started and parked in the open parking lot on the west side of the Electronic Information Engineering College of the Hebei University of Technology. After waiting for about 20 min, the exhaust gas emitted by the vehicle reaches a relatively stable distribution around it. A remote control is used to control the F330 rotorcraft to take off about 5 m from the horizontal direction of the vehicle and 1.8 m from the ground. Then, the UAV flies to the vehicle and carries out hovering sampling around the vehicle. At the same time, the ground station system records the flight track and various gas concentration data of the UAV, as shown in Figure 22.

The whole process of the UAV taking off and returning to the starting point is intercepted, and the result of the rotorcraft flight trajectory is shown in Figure 23a. The blue line denotes the flight trajectory of the UAV; the green diamond, the rotorcraft take-off point of the UAV; the orange diamond, the landing point of the UAV, and the red triangle, the location of the vehicle. The gas monitoring data are preprocessed and the data during the warm-up period are removed and normalized. Changes in various gas concentrations during the entire flight are shown in Figure 23b.

The experimental results show that the concentrations of CO, SO_2_, NO_2_, and CO_2_ are significantly increased near the vehicle position, and the system can still carry out effective gas environment monitoring under the disturbance of UAV airflow.

## 5. Conclusions

The overall hardware design of the monitoring terminal is hereby expounded. In the first part, the selection of sensors is carried out, and the NIDR carbon dioxide sensor, particulate matter sensor and temperature and humidity sensor are finally selected after comparing and analyzing the performance of various sensors. At the same time, the design of each part of the hardware circuit is introduced in detail, and the signal analysis of the driving circuit of the electrochemical sensor is carried out to obtain the original signal output value of each sensor; Then, the cavity design and CFD simulation are conducted to reduce the influence of the airflow disturbance of the UAV and improve the response speed. The simulation results show that the airflow conditions in the response cavity of each small cavity are basically the same in the designed series cavity. The designed buffer cavity can buffer the airflow when the speed of the air pump fluctuates so that the airflow into the response cavity remains relatively stable; finally, the software design is performed. In the second part, taking the CO sensor as an example, the CFD simulation experiment is compared and analyzed. The experimental results show that the designed series type can effectively shorten the response time of the sensor by 7.6 s. Considering the sensitivity of the electrochemical sensor and the difference in baseline drift that leads to the problem of bias in the results, the sensor calibration and temperature compensation are carried out. Based on the linear regression algorithm, the calibration model experiment of the electrochemical sensor is designed and accomplished. The experimental results show that the R^2^ of the model reaches 0.9981, and the fitting effect is rather high. After calibration, the output is much closer to the real gas concentration value. Finally, the design experiment of the segmented temperature compensation method is carried out. The temperature compensation parameters are determined, thereby improving the accuracy of the entire hardware terminal. In the third part, a vehicle exhaust monitoring experiment is designed to verify the feasibility of the gas monitoring terminal, which is proven capable of detecting the emission position of the vehicle exhaust under the interference of the downwash flow of the rotorcraft.

## Figures and Tables

**Figure 1 sensors-22-05350-f001:**
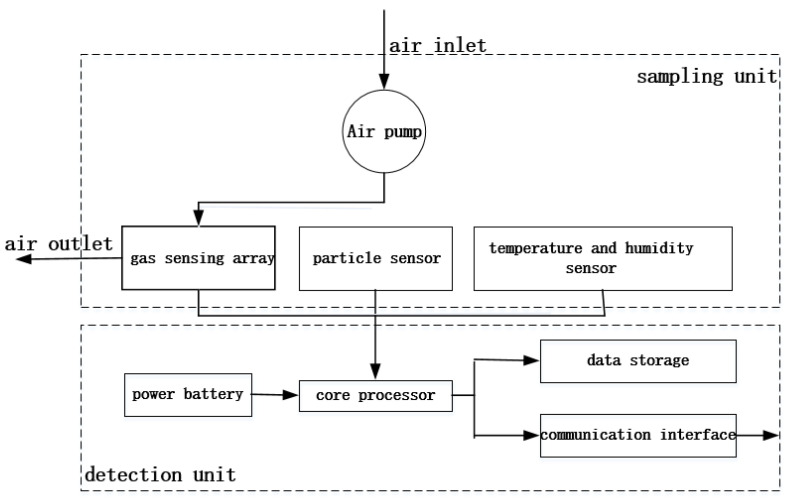
Hardware structure of monitoring terminal.

**Figure 2 sensors-22-05350-f002:**
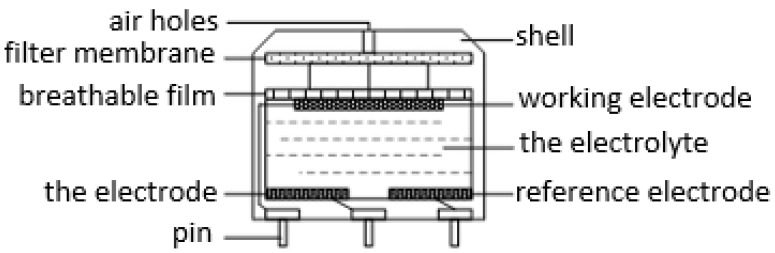
Three electrode electrochemical sensor structure.

**Figure 3 sensors-22-05350-f003:**
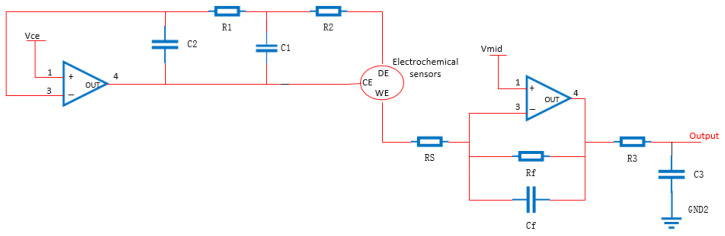
Classic conditioning circuit structure.

**Figure 4 sensors-22-05350-f004:**
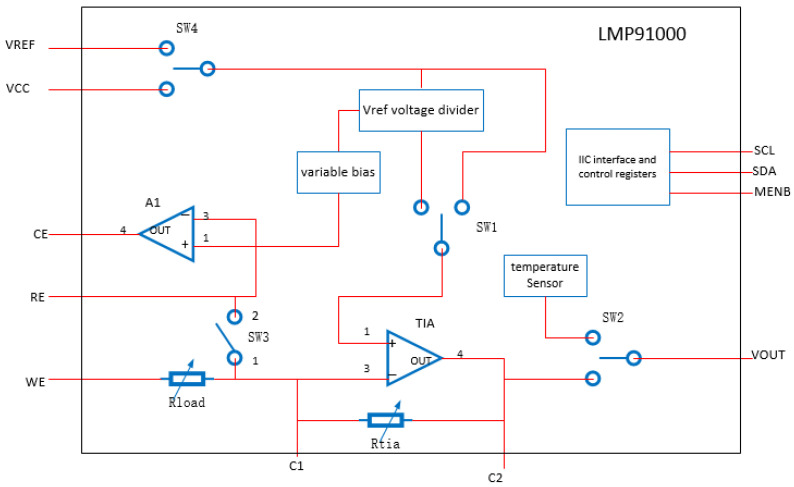
Working principle diagram of LMP91000.

**Figure 5 sensors-22-05350-f005:**
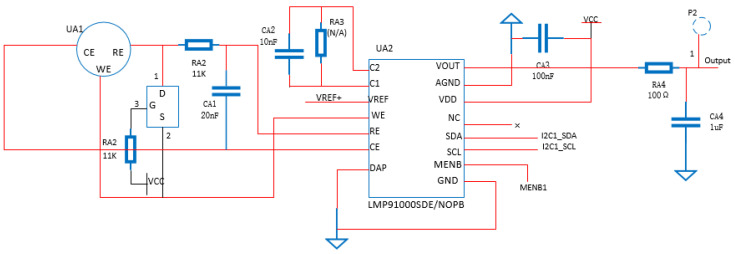
Electrochemical sensor circuit based on LM91000.

**Figure 6 sensors-22-05350-f006:**
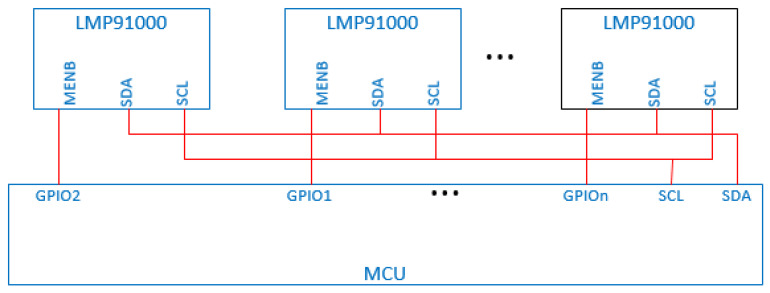
I2C connection mode of multiple LMP91000.

**Figure 7 sensors-22-05350-f007:**
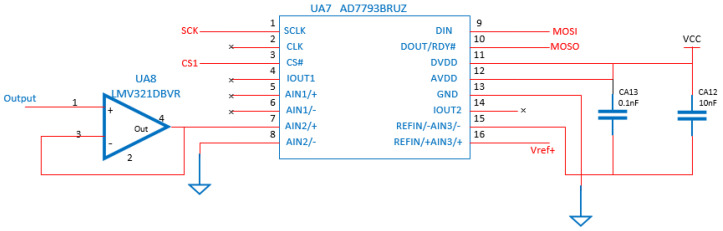
Signal acquisition circuit.

**Figure 8 sensors-22-05350-f008:**
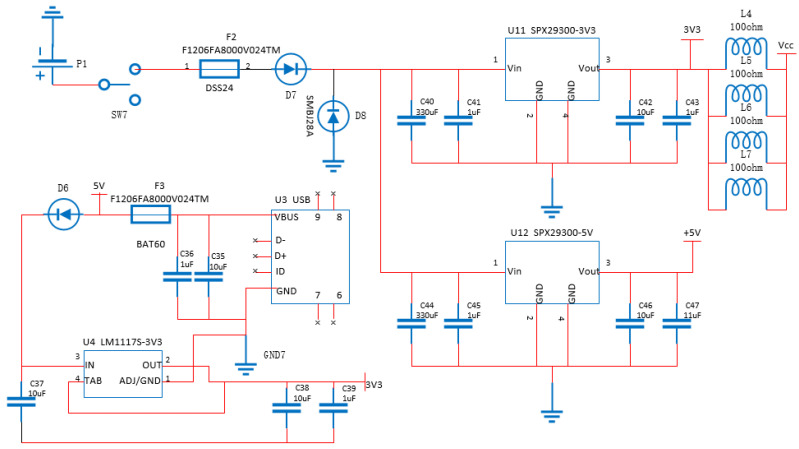
Power supply circuit design.

**Figure 9 sensors-22-05350-f009:**
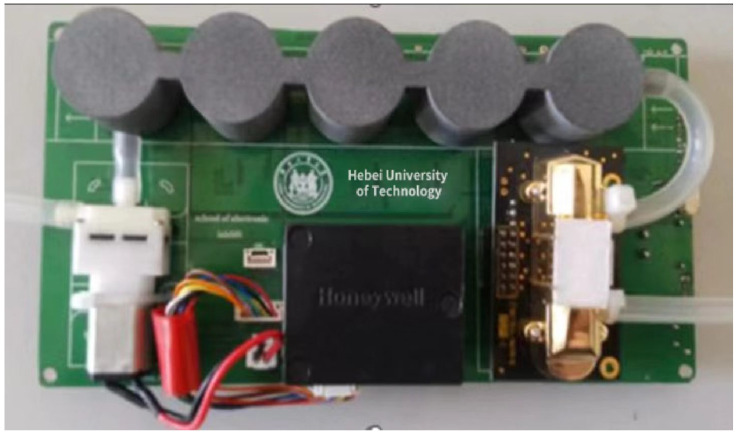
PCB physical picture of monitoring terminal.

**Figure 10 sensors-22-05350-f010:**
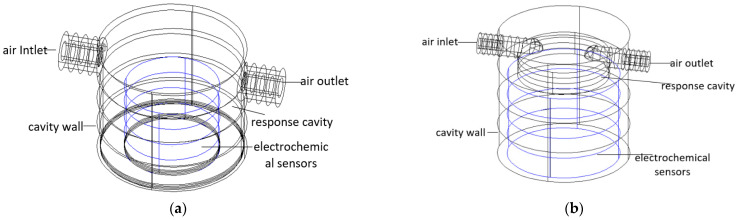
(**a**) Cover cavity structure; (**b**) Cap cavity structure.

**Figure 11 sensors-22-05350-f011:**
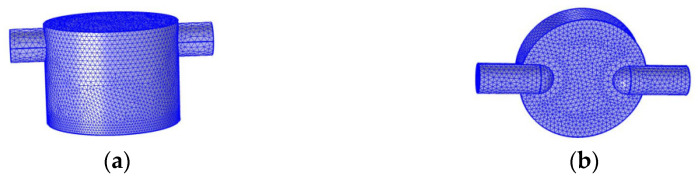
(**a**) Cover cavity airflow field gridding; (**b**) Cap cavity airflow field gridding.

**Figure 12 sensors-22-05350-f012:**
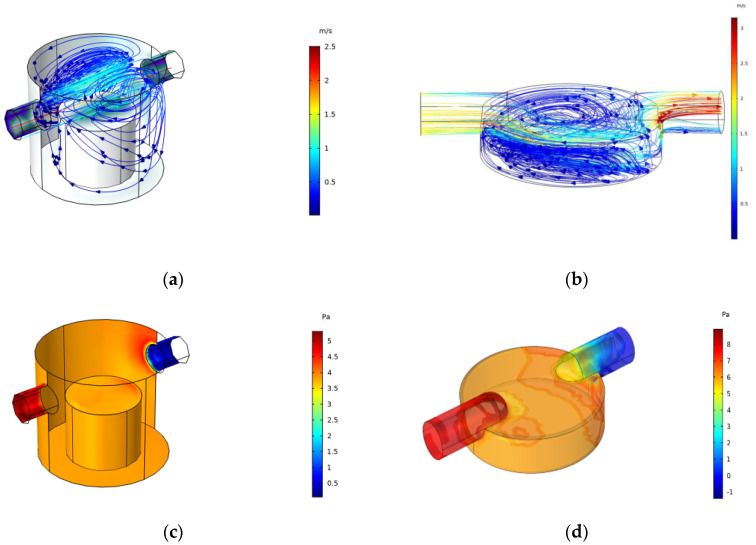
(**a**) Simulation results of velocity field in a cover cavity; (**b**) Simulation results of velocity field in a cap cavity; (**c**) Simulation results of pressure field in a cover cavity; (**d**) Simulation results of pressure field in a cap cavity.

**Figure 13 sensors-22-05350-f013:**
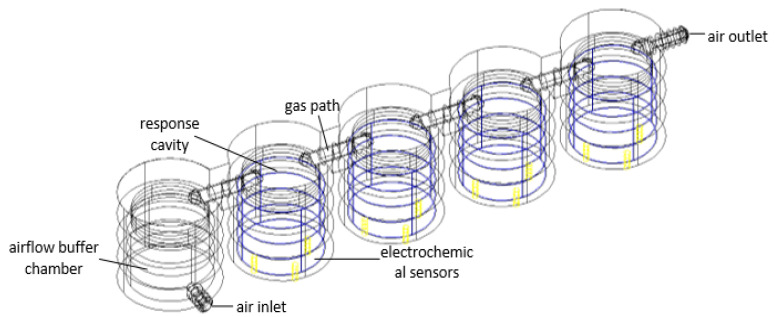
Series cavity structure.

**Figure 14 sensors-22-05350-f014:**
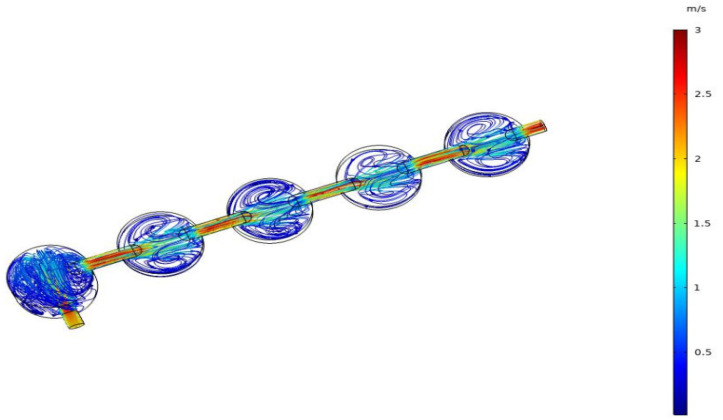
Simulation results of the velocity field in series cavity.

**Figure 15 sensors-22-05350-f015:**
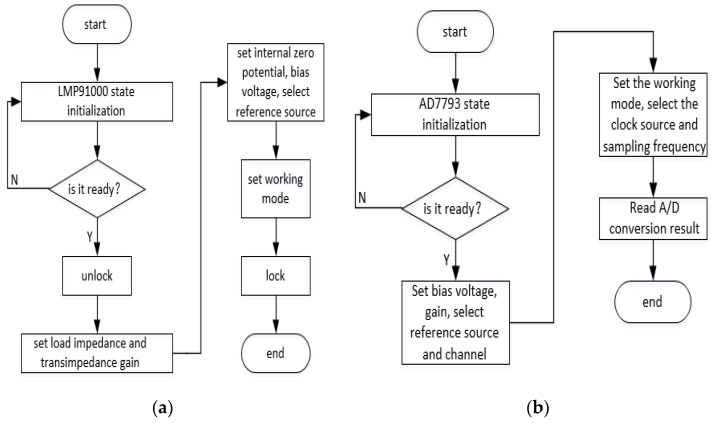
(**a**) LMP91000 program flow; (**b**) AD7793 program flow.

**Figure 16 sensors-22-05350-f016:**
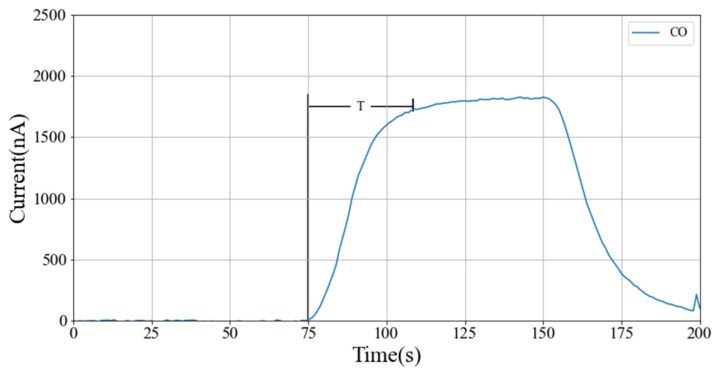
Response results of CO sensors.

**Figure 17 sensors-22-05350-f017:**
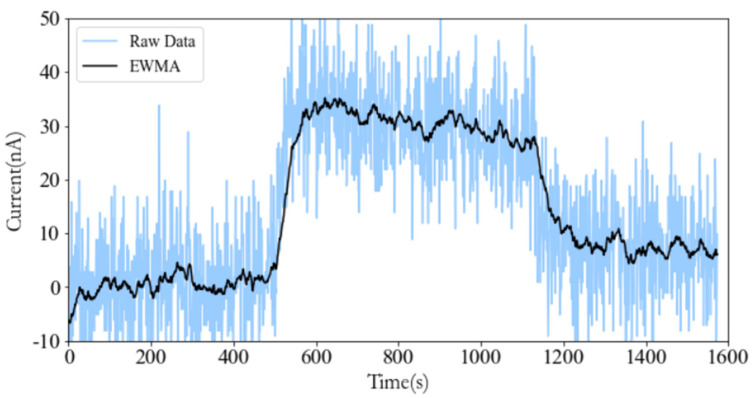
Comparison of before and after EWMA filtering.

**Figure 18 sensors-22-05350-f018:**
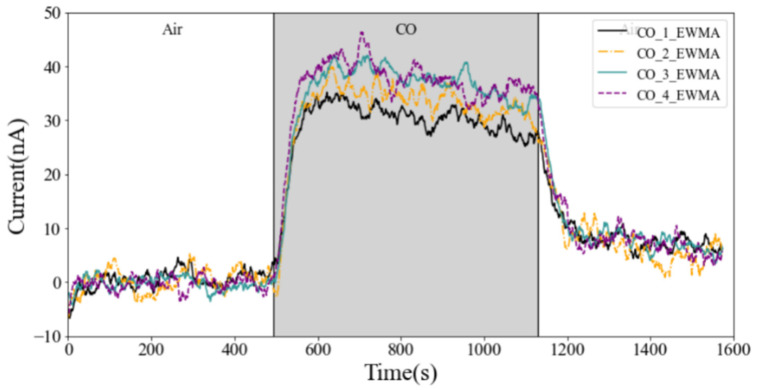
Response results of 4 CO sensors in series cavity.

**Figure 19 sensors-22-05350-f019:**
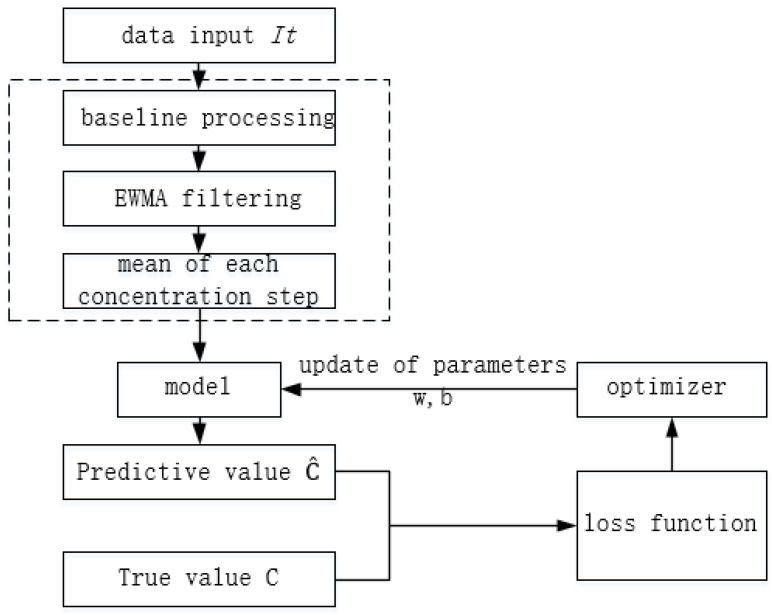
Preprocessing and model training process.

**Figure 20 sensors-22-05350-f020:**
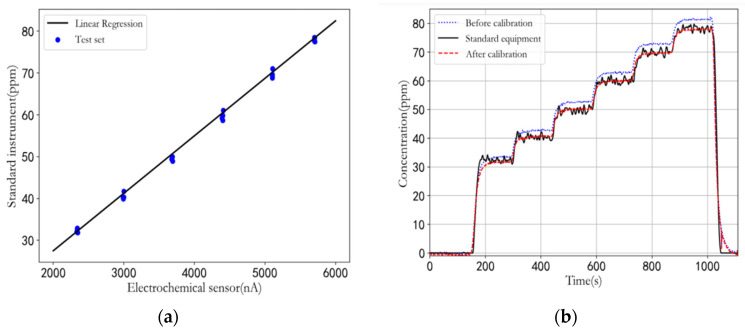
(**a**) Result of linear regression; (**b**) Comparison of linear regression model before and after calibration.

**Figure 21 sensors-22-05350-f021:**
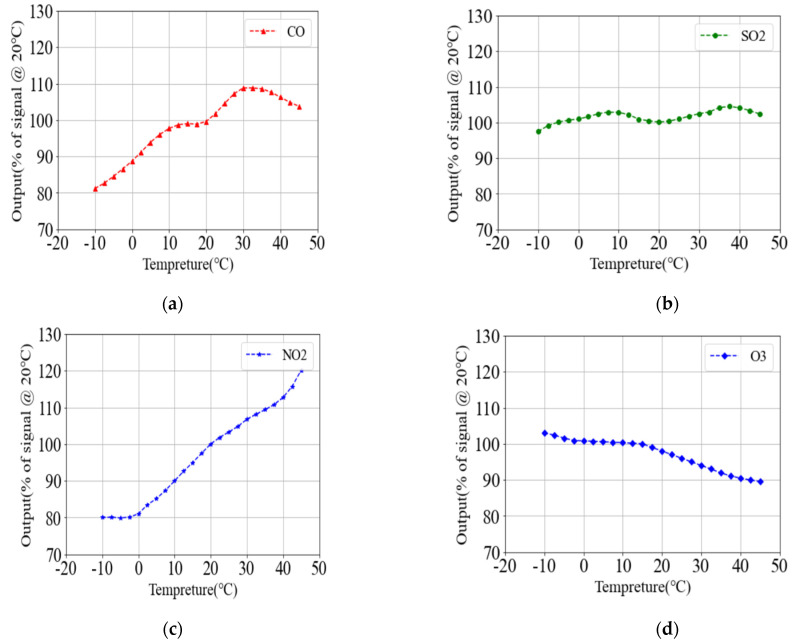
(**a**) CO gas temperature compensation curve; (**b**) SO_2_ gas temperature compensation curve; (**c**) NO_2_ gas temperature compensation curve; (**d**) O_3_ gas temperature compensation curve.

**Figure 22 sensors-22-05350-f022:**
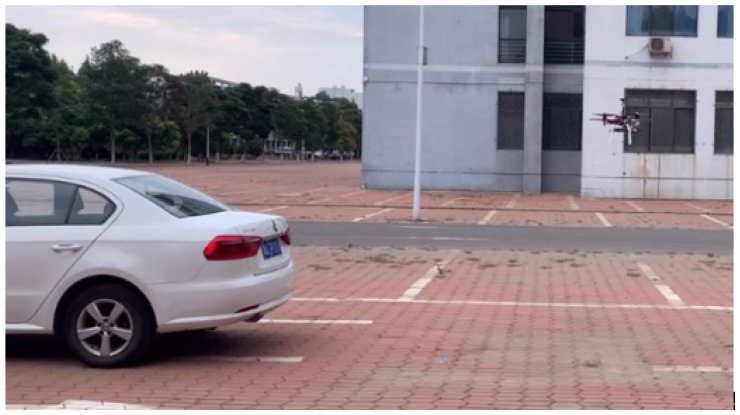
UAV pollution point detection experiment.

**Figure 23 sensors-22-05350-f023:**
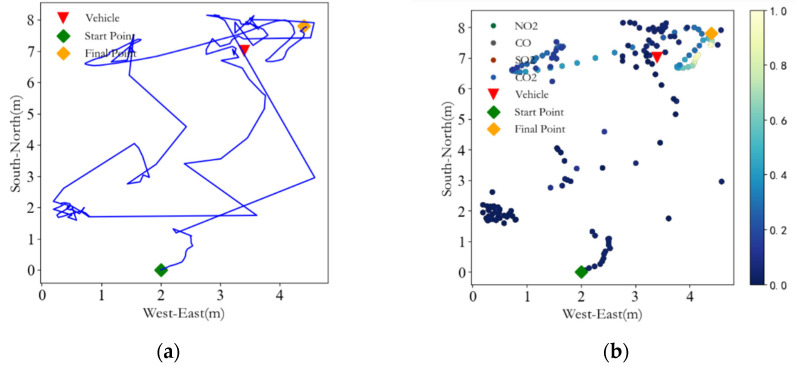
(**a**) Flight trajectory; (**b**) The changes in various gas concentrations.

**Table 1 sensors-22-05350-t001:** Characteristic parameters of 4 kinds of sensors.

Sensitive Gas	Model	Range	Resolution	Sensitivity	Bias	Response Time
CO	4CO-500	0–500 ppm	1 ppm	70 ± 15 nA/ppm	0 mV	≤30 s
SO_2_	4SO_2_-20	0–20 ppm	0.1 ppm	500 ± 100 nA/ppm	0 mV	≤45 s
NO_2_	4NO_2_-20	0–20 ppm	0.1 ppm	600 ± 150 nA/ppm	0 mV	≤30 s
O_3_	4O_3_-10	0–10 ppm	0.05 ppm	850 ± 250 nA/ppm	0 mV	≤45 s

**Table 2 sensors-22-05350-t002:** Comparison of particulate matter sensors.

Parameter Description	HPM 115S0-XXX	HPM 115S0-XXX
manufacturers	Honeywell	ENVIRONMENTAL MONITORING
supply voltage	5VDC	5VDC
PM2.5 effective range	0~1000 µg/m^3^	N/A
PM10 effective range	0~1000 µg/m^3^	0~1000 µg/m^3^
response time	6 s	15 s
resolution	15 µg/m^3^	15 µg/m^3^

**Table 3 sensors-22-05350-t003:** CO_2_ sensors performance comparison.

Parameter Description	S8-0053	T6615-5KF	SCD30	DS-CO_2_-20
manufacturers	Sense Air	Telaire	Sensirion	Panteng Technology
monitoring scope	400~2000 ppm	0~5000 ppm	400~10,000 ppm	400~3000 ppm
monitoring accuracy	±40 ppm	±75 ppm	±30 ppm	±50 ppm
response time	2 s	4 s	2 s	3 s
detection method	diffusion	inhalation	diffusion	diffusion

**Table 4 sensors-22-05350-t004:** Cavity type comparison.

Cavity Type	Cover Cavity	Cap Cavity
boundary structure	simple, few details	The entrance and exit structure is complex
calculation area	larger	less
meshing	coarse mesh	Refine the mesh
grid number	20,146	58,909

**Table 5 sensors-22-05350-t005:** Temperature compensation parameters.

Parameter	CO	SO_2_	NO_2_	O_3_
$a_{1}$	0.809047	0.129048	0.426190	0.119048
$b_{1}$	−0.015428	−0.004571	0.034286	0.076571
$c_{1}$	−0.002133	−0.002667	0.002667	0.004267
$d_{1}$	88.667142	100.967143	92.478571	100.817143
$a_{2}$	1.294312	0.525344	0.082011	−0.085079
$b_{2}$	−0.026984	−0.020508	0.020635	0.004381
$c_{2}$	−0.000740	−0.000830	−0.000296	−0.000178
$d_{2}$	88.166667	100.466667	93.166667	100.9
$a_{3}$	−14.503306	−4.650317	−5.401587	−0.4
$b_{3}$	0.640634	0.185524	0.253333	0
$c_{3}$	−0.008740	−0.002311	−0.003556	0
$d_{3}$	203.32412	137.411905	135.023809	0
$a_{4}$	17.522486	15.297619	3.047884	−5.246825
$b_{4}$	−0.448413	−0.354286	−0.007460	0.116905
$c_{4}$	−0.003704	0.002667	0.000593	−0.000089
$d_{4}$	114.186507	−111.571429	64.873016	171.333

## Data Availability

Not applicable.

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
