# Peer review of "Design of Gas Monitoring Terminal Based on Quadrotor UAV"

_sensors, 2022, doi:10.3390/s22145350_

Round 1
Reviewer 1 Report
1. Lines 1 through 10 of the introduction give too much background, and it is suggested to simplify it.
2. Please refer to standard formats such as travel numbers and the flow chart of Figure 1 and Figure 19.
3. The resolution of Figure 15 is incorrect. The graph has been distorted; the author can consider replacing it with a clearer graph.
4. Please refer to the correct format for modification, such as Figure 1 and Figure 19.
5. What does “T” in Figure. 16 represent, the author also mentioned "T" in Section 3.3. Is the "T" in these two places the same? Please add an explanation.
6. EWMA algorithm is selected as the main algorithm for denoising. The author can compare with other algorithms to get optimal results, such as SMA, WMA, etc.
7. In 3.1, the author mentioned “In the atmospheric environment, the concentration of various polluting gases is low”, please give the specific range.
8. Please specify whether the parameters in the explanation of the formula should be in bold or italic, for example, Eq. 11, Eq. 13, etc.
9. The proposed gas monitoring terminal is used for atmospheric detection in the environment, but in the actual experiment, the author chooses automobile exhaust as the test object. What is the concentration of various gases around the vehicle about 20 minutes after the vehicle starts, and whether it meets the low concentration of various polluting gases in the environment?
10. Is R2 mentioned in the conclusion a determinable coefficient? Please give specific explanation and computational process.
Reviewer 2 Report
This study mainly focused on the overall hardware-designing process of the gas monitoring system based on a quadrotor UAV.
First, various sensors were tested and analyzed. The performance of sensors such as NIDR CO2 sensor, PM sensor, temperature and humidity sensor were carried out and selected. Then, the design of hardware circuit was introduced and signal analysis was performed. In order to reduce the airflow disturbance of the UAV, the cavity was designed by CFD simulation, which is also compared with the experimental data. In addition, aiming at the sensitivity of the electrochemical sensor, sensor calibration was carried out. Finally, this scheme can successfully detect the emission position of the vehicle exhaust under the interference of the downwash flow of the rotorcraft.
Although the proposed issues, the gas monitoring system based on quadrotor UAV may be interesting for readers in this journal, most of all, I supposed the topic covered in this manuscript has too broad and be considered a report rather than a paper. So, I’m against the publication of this manuscript.
Reviewer 3 Report
Research work and real testbed experimentation is very interesting. Also, the topic covered is important to monitor and control the environments and can efficiently help in reducing or least in maintaining the air pollution.
Most of the issues are on the language side.
In abstract, define R2
Define abbreviation used at the first place, rest of the text underneath only abbreviation is fine.
cannot get the meaning, "the fitting degree is very high".
Also, last sentence "The experimental results show that this scheme can successfully detect the exhaust position of the vehicle exhaust under the interference of the downwash flow of the UAV, which proves the reliability and accuracy of the monitoring terminal.", as well is confusing, need to rewrite.
In conclusion:
This article expounds the overall hardware >>This article presents the overall hardware
" By comparing and analyzing the performance of various sensors, NIDR carbon dioxide sensor, particulate matter sensor and temperature and humidity sensor are determined.", hard to understand.
this sentence conveys same idea as the first, "At the same time, the design of each part of the hardware circuit is introduced in detail,", better remove first sentence from conclusion or either.
" and the signal analysis of the driving circuit of the electrochemical sensor is carried out to obtain the original signal output value of each sensor.", very confusing.
"The designed buffer cavity can buffer the airflow when the speed of the air pump fluctuates, so that the airflow into the response cavity remains relatively stable.", again and again same wordings and rest not in flow that why very confusing writeup.
"The CFD simulation experiment is compared and analyzed.", compared with what?
in introduction:
rewrite or combine with previous , "The latter can only monitor air quality in a small area."
"Once the monitoring range is large, it needs to consume a lot of large labor costs. In some special scenarios, such as toxic and harmful gas leakage, emergency personnel cannot intervene, so the effect cannot be achieved", confusing.
". Since the downwash air flow generated by the UAV during the movement will interfere with the final reading of the monitoring terminal, in order to solve this problem, a pump-suction series cavity is designed in this article.", rewrite.
At the end of introduction section, write a paragraph on the rest of paper organization / means how many sections and what they discuss about.
2. Materials and Methods section requires introductory sentence before 2.1.
Section 3. Results as well requires introductory before going to 3.1.
In this condition very to understand your work.
The manuscript should be professionally edited by Elsevier language editing, American Journal Experts, or similar, and while submitting the revised version attach the editing certificate.
Compare you results with any of the recent related published work to justify and show the strength of your work.
Round 2
Reviewer 2 Report
In 2.2.1 Fluid model selection, if commercial code was employed, add it in reference.
If the turbulence model was used, the authors don’t need to explain the fluid model depending on the Reynolds number.
Equation 5 should be expressed in tensors type refer to equation 6.
In this work, the energy equation was not used, so equation 7 should be excluded.
The size of the text in figure 12 should be bigger.
The graph style of the manuscript should be unified.
Reviewer 3 Report
some of the comments are not address appropriately.
Compare you results with any of the recent related published work to justify and show the strength of your work. In case not possible, justify it i a strong manner.
Still the write up is hard to understand. If the readers at global level have issues in understanding your work, will not attract much citations.
The manuscript should be professionally edited by Elsevier language editing, American Journal Experts, or similar, and while submitting the revised version attach the editing certificate.
Cite work from MDPI Sensors journal to show your article relevancy with the journal scope.
Please, attach the article with track in the next submission.
